# Feature Collapse Under Corruption: An Entropy Perspective on Robust Neural Networks

**Vishesh Kumar,** [1]   **Akshay Agarwal** [1]

https://github.com/visheshrajput2408/Dem-HEC.git

## Abstract

Even after decades of advances in neural network training, the inherent robustness challenge remains open. While the sensitivity to adversarial perturbations is understandable given their intentional learning, the most surprising fact is the vulnerability to natural corruptions. Surprisingly, not only is the cause of this inherent vulnerability unknown, but the concern extends beyond traditional CNNs; it also applies to current models, including transformers and large foundation models. For the first time, through this work, we observe that natural corruptions often collapse the network's internal feature space into a high-entropy state, causing predictions to rely on a small subset of fragile features. Inspired by this, we propose a simple yet effective entropy-guided fine-tuning framework, Dem-HEC, that strengthens corruption robustness while maintaining clean accuracy. Our method generates high-entropy samples within a bounded perturbation region and repairs the model using both clean and high-entropy samples. We further combine this objective with distilling knowledge from a teacher snapshot to maintain stable predictions. The proposed Dem-HEC is effective across datasets ranging from small to large-resolution, from pure CNNs to transformers, and to large foundation models, including DinoV3. The proposed approach outperforms the state-of-the-art (SOTA) models not only in improving robustness but also in retaining or boosting clean accuracy.

---

[1]Trustworthy BiometraVision Lab, Indian Institute of Science Education and Research Bhopal, India. Correspondence to: Akshay Agarwal <akagarwal@iiserb.ac.in>.

*Proceedings of the 43$^{rd}$ International Conference on Machine Learning*, Seoul, South Korea. PMLR 306, 2026. Copyright 2026 by the author(s).

## 1. Introduction

The tremendous success of end-to-end deep learning models has led to their deployment across almost every field of vision and on nearly every possible digital device, ranging from laptops to mobile devices. However, still contrary to human vision, these systems are imperfect in handling out-of-distribution (OOD) samples, especially where the samples are affected by natural, also known as common, corruptions (Recht et al., 1806; Azulay & Weiss, 2019; Mitra et al., 2024; Agarwal et al., 2024b; Pedraza et al., 2022). This kind of robustness against OOD images affected by natural corruption is a crucial objective for machine learning and computer vision tasks, in case they genuinely need to be autonomous. In general, imaging accuracy is measured as in-distribution performance, which means a model trained and applied to the same kind of data without any distributional shift. But in practice and unconstrained environments, deep neural networks (DNNs) often encounter data distributions that are different from those used during training. Surprisingly, modelling every form of common corruption is not feasible, and even including them in training can significantly increase computational cost. *Therefore, the robustness must be an inherent part of any network training, because the deployment of models must not be restricted to any environment.* For example, the significant number of steps involved in image acquisition introduces several sources of noise in the images (Agarwal et al., 2020). For example, CMOS sensors are prone to several types of noise, including photon shot noise and amplifier noise, particularly in low-light settings (Bigas et al., 2006). Similarly, transferring or storing images on edge devices requires compression, which can generate image artifacts. Moreover, if the model is truly universal and has no geographical boundaries, it must account for several environmental factors, such as snow and frost.

**Corruption robustness.** Let $C$ denote a set of corruption functions and $f : X \rightarrow Y$ be a classifier trained on samples from a distribution $D$ that does not include any corruptions from $C$. The robustness of $f$ is evaluated by its average performance when classifying corrupted inputs, where the corruptions are drawn from $C$ (Hendrycks & Dietterich,

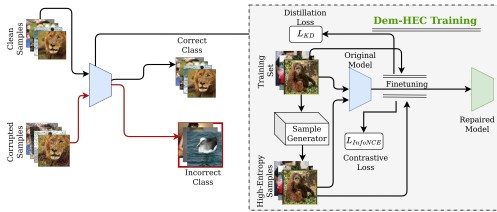

*Figure 1.* An overview of our Dem-HEC framework. The left panel illustrates the traditional approach to handling corrupted samples. The right panel details the training procedure, which combines contrastive learning and knowledge distillation to improve robustness.

2019). Formally, this is expressed as

$$\mathbb{E}_{c \sim C} \; \mathbb{P}_{(x,y) \sim D}\big(f(c(x)) = y\big).$$

Deep neural networks (DNNs) perform worse under distribution shifts where the training and test data distributions differ (Zhou et al., 2024; Kumar & Agarwal, 2026). Before implementing DNNs in the unpredictable and noisy real world, it is essential to assess the consequences of incorrect decisions made by these networks, regardless of the cause, including image corruption. DNNs, including state-of-the-art transformers, remain highly sensitive to noise. When trained only on clean images, these models perform poorly even at noise severity level 1 and degrade further as severity increases from 1 to 5 (Kumar et al., 2025). Similar performance degradation has been observed across different natural distribution shifts (Knoll et al., 2019; Darestani et al., 2021; Dhake & Agarwal, 2024).

Extensive research (Kumar et al., 2025) has benchmarked this vulnerability, revealing that different model architectures exhibit unique sensitivities. This indicates that there is no single "silver bullet" architecture that is universally robust, highlighting the need for methods that can bolster a model's resilience regardless of its design. Moreover, the literature usually deals with specific attacks, leaving the defense in a hollow space. For example, democratic training (Sun et al., 2025) defends against Universal Adversarial Perturbations (UAPs). However, these studies hold some significance and may serve as a basis for future defenses. Henceforth, inspired by this democratic training approach, we address the challenge of robustness against multiple natural corruptions to achieve unified defense. We hypothesise that, unlike UAPs, which induce feature dominance and low entropy, natural corruptions introduce ambiguity and uncertainty, which can be modelled by an increase in feature space entropy as discussed in subsection 3.1. Further, the high-entropy samples help reduce the gap between the training and test distributions by reducing reliance on spurious correlations and discouraging memorization. Therefore, we propose a novel fine-tuning framework, Dem-HEC (Democratic High-Entropy samples for Corruption robustness), as

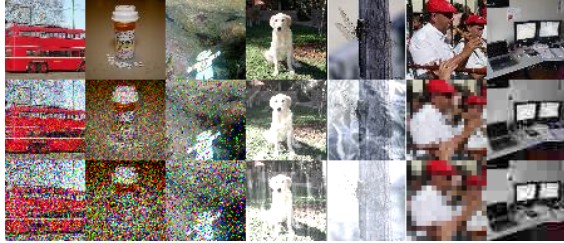

*Figure 2.* Visual examples of the seven common corruptions used in our evaluation. The first row displays the original clean images. The second and third rows show the corresponding corrupted images at severity levels 3 and 5, respectively. The corruptions, from left to right, are: Gaussian noise, shot noise, impulse noise, snow, frost, pixelate, and JPEG compression.

described in Figure 1 that takes the opposite approach to Democratic Training (Sun et al., 2025). Instead of suppressing dominant features, Dem-HEC encourages the model to learn invariant representations by training it on challenging high-entropy samples. These samples are generated via gradient ascent on the entropy of the model's feature space, pushing the model to make stable predictions even when feature activations are maximally uncertain. To achieve this, we introduce a composite loss function that combines four key objectives: (1) standard cross-entropy on clean images to maintain baseline accuracy, (2) cross-entropy on our generated high-entropy samples to learn robust features, (3) a contrastive loss to ensure that the representations of clean images and their high-entropy counterparts remain semantically similar, and (4) knowledge distillation to prevent the model from catastrophically forgetting the knowledge of the original pre-trained network. We demonstrate, through extensive experiments on CIFAR10, CIFAR100, Tiny-ImageNet, a Subset of ImageNet (ImageNette), and full ImageNet-1K with various backbones (ResNet, ViT, RepVGG, DINOv3), that Dem-HEC significantly enhances robustness against a wide range of common corruptions and their severities, often outperforming models trained on clean data alone.

## 2. Notation and Definitions

### 2.1. Common Corruption

In this work, we focus on seven widely recognized corruption types that reflect real-world degradations encountered in image acquisition, transmission, and storage. The first category consists of additive noise corruptions: *Gaussian noise*, *Shot noise*, and *Impulse noise*. The second category involves environmental corruptions: *Snow corruption* and *Frost corruption*. Finally, we consider digital corruptions, which are consequences of post-capture transformations: *Pixelation* and *JPEG compression*. Together, these seven corruption types cover a broad range of sensor-level, environmental, and digital artifacts, providing a comprehensive

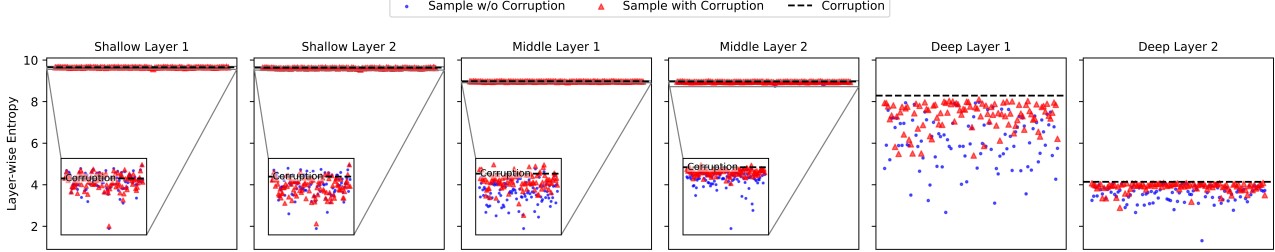

*Figure 3.* Layer-wise entropy for a ResNet-20 on CIFAR10 with pixelation corruption with severity 5. Entropy clearly separates clean (blue) and corrupted (red) samples in deep layers, while remaining uniformly high for both in shallow and middle layers.

testbed for evaluating the robustness of deep neural networks against corruption. Moreover, for comprehensiveness, each corruption has been applied with multiple severities reflecting mild (S1), medium (S3), and high (S5) severity. The corresponding severity parameter has been inspired by the work of (Hendrycks & Dietterich, 2019). Figure 2 shows the challenge that the proposed research is handling by tackling the loss of visual cues at high severities, which in turn demonstrates the strength and impact of the proposed research. The details about these corruptions are provided in the appendix A.1.

## 2.2. Evaluation Metrics

**Corrupted Accuracy (CAcc.):** This metric measures the accuracy of corrupted examples (where $y_x$ represents the label of sample $x$):

$$CAcc. = \sum_{x \in X} \frac{|f(x + \delta) = y_x|}{|X|} \tag{1}$$

## 2.3. Entropy of a Neural Network

In information theory, Shannon entropy is a fundamental measure that quantifies the average level of uncertainty or information contained in the outcomes of a random variable. First introduced by Claude Shannon (Shannon, 1948), this concept captures how much "surprise" or unpredictability is associated with a probabilistic system. Formally, let $v$ be a random variable that can take values from a set $V$ with an associated probability distribution $p : V \rightarrow [0, 1]$. The Shannon entropy of $v$ is expressed as:

$$H(v) = -\sum_{v \in V} p(v) \log p(v), \tag{2}$$

where the summation is taken over all possible values of $v$.

Entropy has been widely adopted in neural networks to characterize the uncertainty in their internal representations or predictions. Prior work has proposed various strategies for estimating neural entropy at different levels of abstraction (Grandvalet & Bengio, 2004; Shwartz-Ziv & Tishby, 2022; Agarwal et al., 2024a). In this study, we focus on computing *layer-wise entropy* to investigate how common corruptions

**Algorithm 1** HIGH-ENTROPY SAMPLE GENERATOR

**Require:** Input $x$, radius $\epsilon$, step size $\eta$, steps $T_{\text{he}}$
1: Initialize $x^{(0)} \leftarrow \text{clip}(x + \mathcal{U}(-\epsilon, \epsilon))$ (optional random start)
2: **for** $t = 0 \rightarrow T_{\text{he}} - 1$ **do**
3:   Compute gradient $\boldsymbol{g}^{(t)} \leftarrow \nabla_{x^{(t)}} H(\text{softmax}(f(x^{(t)})))$
4:   Ascent step $x^{(t+1)} \leftarrow x^{(t)} + \eta \cdot \text{sign}(\boldsymbol{g}^{(t)})$
5:   Project $x^{(t+1)} \leftarrow \Pi_{\mathcal{B}_\epsilon(x)}(x^{(t+1)})$ and clip to $[0, 1]$
6: **end for**
7: **return** $x' \leftarrow x^{(T_{\text{he}})}$

alter the network's internal feature distributions. A detailed description of this formulation is presented in subsection 3.1.1.

## 2.4. Problem Formulation: Common Corruption Robustness

Let $F$ denote a neural network classifier, $x \in \mathbb{R}^{H \times W \times C}$ be a clean input with ground-truth label $y$. Consider a family of corruption operators

$$\mathcal{G} = \{g_c(\cdot, s) \mid c \in \mathcal{C}, \ s \in \{1, 3, 5\}\},$$

where each $g_c : \mathbb{R}^{H \times W \times C} \rightarrow \mathbb{R}^{H \times W \times C}$ represents a corruption of type $c$ (e.g., Gaussian noise, shot noise, impulse noise, snow, frost, pixelation, JPEG compression) applied with severity level $s$.

The *common corruption robustness problem* is to design a defense strategy such that the network's predictions remain reliable under these corruptions:

$$\arg\max F(x) = y$$
$$\implies \arg\max F(g_c(x, s)) = y,$$
$$\forall c \in \mathcal{C}, \ s \in \{1, \dots, 5\}. \tag{3}$$

At the same time, the defense must preserve the classifier's performance on clean data, i.e., the accuracy on uncorrupted inputs $x$ should remain close to that of the original network.

## 3. Proposed Robustness Approach

To investigate how natural corruptions affect model behavior, we conduct a systematic analysis through the lens of entropy. Specifically, we examine the *layer-wise entropy* of a given network when processing both clean and corrupted inputs.

### 3.1. Role of Image Corruption on Network Entropy

To understand how natural corruptions influence the behavior of a trained neural network, we conduct an empirical study on the *layer-wise entropy* of the model as follows: **Step 1.** Given a pretrained neural network, we collect a set of clean test samples and compute their layer-wise entropy as defined in Equation (4). **Step 2.** Apply different natural corruptions (e.g., Gaussian noise, shot noise, impulse noise, snow, frost, pixelation, JPEG compression) with varying severity levels to the same set of samples. **Step 3.** Compute and compare the layer-wise entropy of clean inputs versus corrupted inputs. **Step 4.** Analyze the evolution of entropy across shallow, middle, and deep layers to understand how corruptions alter uncertainty.

#### 3.1.1. ENTROPY MEASUREMENT

We begin by defining entropy in our setting. Consider a neural network $F$ consisting of $n$ layers. Each layer $l$ can be treated as a random variable characterized by its input $x_l$ and output $x_{l+1}$. For a layer with $d_l$ neurons, given input

$$x_l = \{x_l^0, x_l^1, \ldots, x_l^{d_l-1}\},$$

its activations are computed as

$$\chi_l = \sigma(W_l x_l + b_l),$$

where $W_l$ and $b_l$ denote the weights and biases of layer $l$, and $\sigma(\cdot)$ is its activation function. The normalized activation distribution is obtained via

$$p_l = \text{softmax}(\chi_l).$$

Finally, the *layer-wise entropy* is defined as

$$H_l = -\sum_{k=0}^{d_l-1} p_l(k) \log p_l(k). \tag{4}$$

Intuitively, we treat the activation probability $p_l(k)$ of neuron $k$ as the likelihood of it being active, and compute the Shannon entropy over all neurons. A higher entropy $H_l$ indicates greater uncertainty or feature diversity, while lower entropy reflects higher certainty or dominance of a small subset of neurons. Under natural corruptions, we often observe an abnormal increase in entropy, suggesting that corrupted inputs cause the network to rely on spurious features rather than balanced feature representations.

As illustrated in Figure 3, before applying the proposed Dem-HEC, at shallow layers, the entropy distributions of clean and corrupted inputs are close to each other, indicating that early convolutional features are relatively stable. However, as inputs propagate through middle and deeper layers, corrupted samples consistently exhibit **higher entropy** than their clean counterparts. This effect becomes more pronounced at deeper layers, where natural corruptions induce substantial ambiguity in the learned representations. These findings suggest that, unlike UAPs, which inject dominant features and reduce entropy, natural corruptions increase entropy by dispersing feature activations, thereby making the model less confident about its predictions. In other words, corruption distorts discriminative cues, forcing the network to rely on noisy or occluded signals, which increases uncertainty. Our analysis thus highlights a key contrast: *UAPs enforce artificial certainty (low entropy), while natural corruptions degrade representation quality and amplify uncertainty (high entropy).*

Motivated by these findings, we propose **Dem-HEC**, an entropy-guided training framework that enhances model robustness against natural corruptions by encouraging balanced feature representations.

### 3.2. Proposed Dem-HEC

To mitigate the effects of natural corruptions on neural networks, we propose Dem-HEC, a general framework applicable to different architectures (e.g., CNNs such as ResNet-18/56, RepVGG-A0/A2, or Transformers such as ViT and Large) and datasets (CIFAR10, CIFAR100, Tiny ImageNet). Unlike existing defenses against universal adversarial perturbations (UAPs), which focus on reducing overconfident, low-entropy activations, our method explicitly accounts for the opposite phenomenon: natural corruptions tend to induce high-entropy predictions (greater uncertainty). Dem-HEC therefore regularizes networks to handle corrupted high-entropy samples while maintaining higher accuracy on clean data.

#### 3.2.1. BACKBONE AND PROBLEM SETUP

Let $f(\cdot; \theta)$ be a pretrained classifier with parameters $\theta$. Given an input image $x \in \mathbb{R}^{H \times W \times C}$ and label $y \in \{1, \ldots, K\}$, the model produces logits $z = f(x; \theta)$ and predictive distribution

$$p(y \mid x) = \text{softmax}(z). \tag{5}$$

The standard cross-entropy loss is

$$\mathcal{L}_{\text{CE}}(f(x; \theta), y) = -\log p(y \mid x). \tag{6}$$

We adopt *partial fine-tuning* (freeze early layers, update higher blocks and head) to retain general features while adapting to corruption robustness.

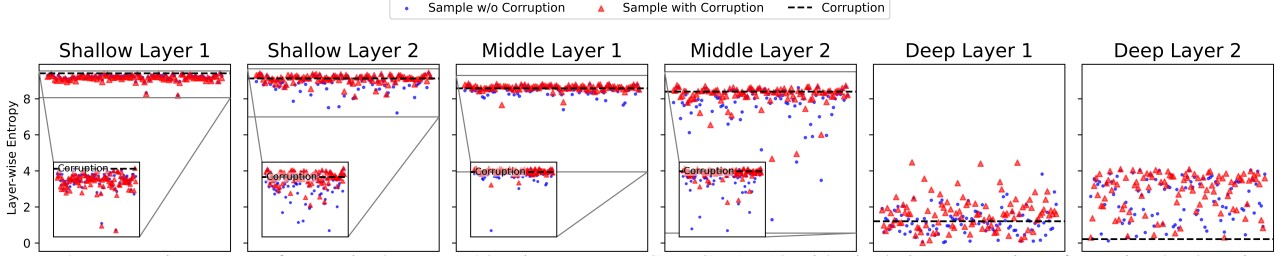

*Figure 4.* Layer-wise entropy for repaired ResNet-20 using Dem-HEC on CIFAR10 with pixelation corruption of severity 5. There is no clear separation of clean (blue) and corrupted (red) samples in deep layers.

#### 3.2.2. HIGH-ENTROPY SAMPLE GENERATION

Natural corruptions typically increase predictive uncertainty in deep layers. We simulate this training signal by synthesizing a *high-entropy* variant $x'$ of $x$ via constrained entropy maximization shown in Algorithm 1.

Let the Shannon entropy of the model output be

$$H(p(\cdot \mid x)) = -\sum_{k=1}^{K} p_k(x) \log(p_k(x) + \varepsilon_0), \quad (7)$$

with a small $\varepsilon_0 > 0$ for numerical stability. We solve

$$x' = \arg \max_{\|x'-x\|_\infty \leq \epsilon} H(p(\cdot \mid x')), \quad (8)$$

using $T$ steps of Projected Gradient Ascent (PGA):

$$x^{(t+1)} = \Pi_{\mathcal{B}_\epsilon(x)} \Big( x^{(t)} + \eta \cdot \mathrm{sign}\Big( \nabla_{x^{(t)}} H\Big( p(\cdot \mid x^{(t)}) \Big) \Big) \Big), \quad (9)$$

where $\eta$ is the step size and $\Pi_{\mathcal{B}_\epsilon(x)}$ projects onto the $\ell_\infty$ ball of radius $\epsilon$ around $x$ (and to the valid pixel range).

#### 3.2.3. CONTRASTIVE REPRESENTATION ALIGNMENT

Let $g(\cdot; \theta)$ be a representation extractor (e.g., penultimate layer), and define $\boldsymbol{v} = \mathrm{norm}(g(x; \theta))$ and $\boldsymbol{v}' = \mathrm{norm}(g(x'; \theta))$, with $\mathrm{norm}(\cdot)$ denoting $\ell_2$-normalization. For a batch of size $B$, $\{\boldsymbol{v}_i\}_{i=1}^B$ and $\{\boldsymbol{v}_i'\}_{i=1}^B$, the symmetric InfoNCE loss is

$$\mathcal{L}_{\mathrm{InfoNCE}} = -\frac{1}{2B} \sum_{i=1}^{B} \Bigg[ \log \frac{\exp(\mathrm{sim}(\boldsymbol{v}_i, \boldsymbol{v}_i')/\tau)}{\sum_{j=1}^{B} \exp(\mathrm{sim}(\boldsymbol{v}_i, \boldsymbol{v}_j')/\tau)} + \log \frac{\exp(\mathrm{sim}(\boldsymbol{v}_i', \boldsymbol{v}_i)/\tau)}{\sum_{j=1}^{B} \exp(\mathrm{sim}(\boldsymbol{v}_i', \boldsymbol{v}_j)/\tau)} \Bigg], \quad (10)$$

where $\mathrm{sim}(\boldsymbol{u}, \boldsymbol{v}) = \boldsymbol{u}^\top \boldsymbol{v}$ and $\tau > 0$ is a temperature.

#### 3.2.4. KNOWLEDGE DISTILLATION FOR CLEAN-DATA STABILITY

To avoid forgetting on clean inputs, we distil from a frozen teacher $f(\cdot; \theta_T)$ into the student $f(\cdot; \theta_S)$ using softened logits:

$$\mathcal{L}_{\mathrm{KD}} = T^2 \cdot \mathrm{KL}\big(\sigma(z_S/T) \,\|\, \sigma(z_T/T)\big), \quad (11)$$

where $z_S = f(x; \theta_S)$, $z_T = f(x; \theta_T)$, $\sigma$ is softmax, and $T > 0$ is the distillation temperature.

#### 3.2.5. TOTAL OBJECTIVE

The complete Dem-HEC loss (per minibatch) combines clean and high-entropy CE terms, contrastive alignment, and KD follows Algorithm 2:

$$\mathcal{L}_{\mathrm{total}} = (1-\alpha)\,\mathcal{L}_{\mathrm{CE}}(x, y) + \alpha\,\mathcal{L}_{\mathrm{CE}}(x', y) + \lambda_C\,\mathcal{L}_{\mathrm{InfoNCE}} + \lambda_{\mathrm{KD}}\,\mathcal{L}_{\mathrm{KD}}. \quad (12)$$

with trade-off coefficients $\alpha \in [0, 1]$, $\lambda_C \geq 0$, and $\lambda_{\mathrm{KD}} \geq 0$.

Figure 4 shows the entropy distribution after training with our high-entropy samples, where clean and corrupted inputs now fall within the same entropy range across all layers. This demonstrates that the generated high-entropy samples successfully reproduce the feature-space uncertainty patterns induced by real natural corruptions, while still preserving semantic structure. These results confirm that the high-entropy samples used in Dem-HEC are consistent with actual corruption behavior and effectively guide the model toward stable, corruption-robust representations.

## 4. Experimental Setup

### 4.1. Datasets and Models

In our experiments, we evaluate the proposed Dem-HEC framework on widely used benchmark datasets containing low to high-resolution images: CIFAR10 (32×32) (Krizhevsky, 2009), CIFAR100 (32×32) (Krizhevsky, 2009), Tiny-ImageNet (64×64) (or referred to as ImageNet200), a 10-class ImageNet subset (referred to as ImageNette (224×224)) (Howard), and ImageNet-1K (224×224). To assess our method across a range of model complexities, we select architectures with diverse parameter counts. For CIFAR10 and CIFAR100, we adopt architectures: ResNet-20 (0.27M params), ResNet-56 (0.66M

**Algorithm 2** Dem-HEC Training (architecture- and dataset-agnostic)

**Require:** Pretrained model $f(\cdot;\theta)$; teacher copy $f(\cdot;\theta_T)$ (frozen); hyperparameters $\alpha$, $\lambda_C$, $\lambda_{KD}$, temperature $T$; PGA steps $T_{he}$, step size $\eta$, radius $\epsilon$.

1: **for** epoch $= 1, \ldots, E$ **do**
2:     **for** minibatch $\mathcal{B} = \{(x_i, y_i)\}_{i=1}^{B}$ **do**
3:         **High-entropy samples:** for each $x_i$, compute $x'_i \leftarrow \text{HE\_GENERATE}(x_i; \epsilon, \eta, T_{he})$
4:         **Forward:** obtain logits $z_i = f(x_i; \theta)$ and $z'_i = f(x'_i; \theta)$
5:         **Embeddings:** $v_i = \text{norm}(g(x_i; \theta))$, $v'_i = \text{norm}(g(x'_i; \theta))$
6:         **Teacher logits (clean):** $z_i^{(T)} = f(x_i; \theta_T)$
7:         **Losses:**

$$\mathcal{L}_{CE}^{clean} = \frac{1}{B}\sum_i -\log \text{softmax}(z_i)[y_i],$$

$$\mathcal{L}_{CE}^{he} = \frac{1}{B}\sum_i -\log \text{softmax}(z'_i)[y_i],$$

$$\mathcal{L}_{InfoNCE} \text{ from } \{v_i\}, \{v'_i\},$$

$$\mathcal{L}_{KD} = \frac{T^2}{B}\sum_i \text{KL}\big(\sigma(z_i/T)\,\|\,\sigma(z_i^{(T)}/T)\big)$$

8:         **Total loss:**
        $\mathcal{L}_{total} = (1-\alpha)\,\mathcal{L}_{CE}^{clean} + \alpha\,\mathcal{L}_{CE}^{he}$
9:                 $+ \lambda_C\,\mathcal{L}_{InfoNCE} + \lambda_{KD}\,\mathcal{L}_{KD}$
10:         **Update:** $\theta \leftarrow \theta - \eta_{opt}\nabla_\theta \mathcal{L}_{total}$
11:     **end for**
12: **end for**

params), RepVGG-A0 (489.08M params), and RepVGG-A2 (1850.1M params). For Tiny-ImageNet, we employ three diverse backbones: ResNet-18, ResNet-50, and a large-scale Vision Transformer (ViT-L) with 304M parameters. This selection of models allows us to test the scalability and generalizability of our method. Apart from that, for CIFAR-100, we adopted a large DINOv3 model. Through the large-scale experiments, we aim to demonstrate that the proposed algorithm is architecture-agnostic. When applying Dem-HEC, we compute entropy primarily at the final pooling or dense layer, as the impact of common corruptions on layer-wise entropy becomes most pronounced in deeper layers, consistent with the analysis presented in Figure 3. The implementation details are given in the appendix A.4.

## 5. Results and Analysis

To validate the effectiveness of our proposed Dem-HEC framework, we conducted a comprehensive evaluation across diverse datasets, ranging from low- to high-resolution images, and using different model architectures. We assess

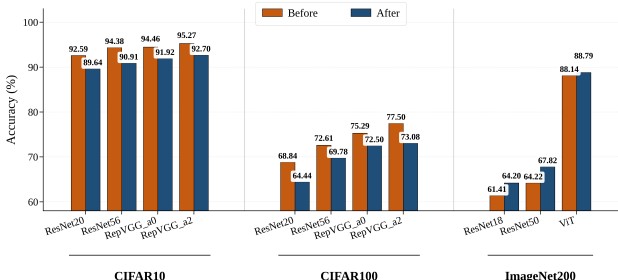

*Figure 5.* Clean accuracy of models on CIFAR10, CIFAR100, and Tiny-ImageNet (or ImageNet200). The comparison shows performance before and after applying Dem-HEC, illustrating that the original accuracy on uncorrupted data is maintained across all architectures. While on a small scale, a marginal drop has been observed, at high resolution, the proposed approach improves performance on clean images.

*Table 1.* Corruption Accuracy (CAcc.) on CIFAR10-C, comparing performance before and after applying Dem-HEC. Our method yields significant robustness gains across all models, particularly for noise-based corruptions and at higher severity levels (S3 and S5).

| Backbone | \multicolumn{6}{ResNet20} | | | | | | \multicolumn{6}{ResNet56} | | | | | |
|---|---|---|---|---|---|---|---|---|---|---|---|---|
| Severity | S1 | | S3 | | S5 | | S1 | | S3 | | S5 | |
| Corruption | Before | After | Before | After | Before | After | Before | After | Before | After | Before | After |
| Gaussian | 71.37 | **88.16** | 30.17 | **74.18** | 21.23 | **61.47** | 75.71 | **89.98** | 37.02 | **78.04** | 25.83 | **66.97** |
| Shot | 80.92 | **88.90** | 43.46 | **79.16** | 25.86 | **64.00** | 83.98 | **90.60** | 50.49 | **82.44** | 31.37 | **68.47** |
| Impulse | 79.90 | **86.09** | 58.16 | **73.65** | 22.95 | **39.82** | 83.17 | **87.00** | 60.66 | **74.71** | 22.70 | **43.35** |
| Snow | 85.58 | **86.94** | 77.12 | **80.08** | 68.33 | **75.81** | 88.02 | **88.64** | 81.20 | **82.59** | 74.35 | **78.39** |
| Frost | 86.59 | **87.62** | 68.99 | **78.36** | 55.55 | **70.86** | 89.35 | **89.22** | 74.84 | **81.92** | 62.33 | **75.61** |
| Pixelate | 88.89 | 88.72 | 74.97 | **86.59** | 39.85 | **73.29** | 91.53 | **90.09** | 80.40 | **88.87** | 44.73 | **78.53** |
| JPEG | 82.94 | **87.72** | 74.88 | **85.29** | 68.28 | **83.29** | 85.25 | **89.32** | 77.28 | **87.12** | 71.20 | **85.43** |
| Backbone | \multicolumn{6}{RepVGG_a0} | | | | | | \multicolumn{6}{RepVGG_a2} | | | | | |
| Severity | S1 | | S3 | | S5 | | S1 | | S3 | | S5 | |
| Corruption | Before | After | Before | After | Before | After | Before | After | Before | After | Before | After |
| Gaussian | 71.93 | **90.99** | 20.99 | **79.69** | 14.37 | **69.29** | 77.22 | **91.51** | 30.34 | **80.95** | 19.04 | **71.42** |
| Shot | 82.89 | **91.56** | 38.23 | **84.17** | 19.30 | **71.40** | 86.52 | **92.36** | 50.15 | **85.40** | 27.65 | **74.01** |
| Impulse | 84.22 | **89.39** | 60.34 | **80.44** | 16.08 | **50.10** | 82.52 | **90.04** | 58.85 | **81.91** | 21.63 | **54.42** |
| Snow | 89.16 | **89.56** | 82.70 | **83.63** | 77.06 | **80.40** | 89.19 | **90.55** | 83.82 | **84.90** | 77.34 | **80.79** |
| Frost | 90.79 | **90.62** | 77.67 | **84.35** | 66.18 | **79.46** | 91.58 | 91.28 | 79.71 | **84.89** | 68.94 | **79.86** |
| Pixelate | 92.78 | **91.14** | 85.86 | **89.19** | 50.31 | **77.95** | 93.20 | 92.04 | 85.58 | **90.16** | 50.27 | **80.54** |
| JPEG | 87.19 | **90.37** | 79.90 | **87.85** | 74.49 | **86.14** | 87.87 | **91.02** | 80.98 | **88.77** | 75.26 | **86.70** |

performance on both clean data and data corrupted by several common types at varying severities.

### 5.1. Performance on Clean Data

A crucial requirement for any robustness enhancement technique is the preservation of performance on uncorrupted (clean) data. Figure 5 illustrates the clean accuracy before and after applying Dem-HEC. The results show that, while on the small-scale datasets (CIFAR10 and CIFAR100), the proposed model exhibits slightly lower performance (in the range 2.5-4.4%) than the base models (though not always), it shows better performance on the large-scale dataset (Tiny ImageNet). For instance, the ResNet56 model on CIFAR100 decreases from 72.61% to 69.78%, while an approximately similar-sized network, i.e., ResNet50, increases from 64.22% to 67.82% on the ImageNet200 dataset. The network's robustness in handling large-scale datasets demonstrates that the proposed approach is scalable to high-resolution images.

*Table 2.* Corruption Accuracy (CAcc.) on Tiny-ImageNet-C, comparing performance before and after applying Dem-HEC. Our method yields significant robustness gains across all models, particularly for noise-based corruptions and at higher severity levels (S3 and S5).

| Backbone | ResNet18 | | | | | | ResNet50 | | | | | | ViT | | | | | |
|---|---|---|---|---|---|---|---|---|---|---|---|---|---|---|---|---|---|---|
| Severity | S1 | | S3 | | S5 | | S1 | | S3 | | S5 | | S1 | | S3 | | S5 | |
| Corruption | Before | After | Before | After | Before | After | Before | After | Before | After | Before | After | Before | After | Before | After | Before | After |
| Gaussian | 46.08 | **54.36** | 16.30 | **24.95** | 7.38 | **10.58** | 49.85 | **57.01** | 14.83 | **27.43** | 5.32 | **12.73** | 79.88 | **80.45** | 54.61 | **59.72** | 34.42 | **39.40** |
| Shot | 45.19 | **53.62** | 24.90 | **34.83** | 9.36 | **13.80** | 49.50 | **55.89** | 25.81 | **36.85** | 7.23 | **15.57** | 80.16 | 80.14 | 65.99 | **68.52** | 39.33 | **43.19** |
| Impulse | 45.76 | **50.01** | 20.34 | **27.33** | 5.66 | **6.63** | 49.26 | **52.67** | 19.18 | **32.08** | 4.49 | **9.33** | 78.77 | 78.65 | 63.91 | 62.04 | 31.80 | 28.80 |
| Snow | 42.61 | **47.97** | 27.28 | **33.36** | 17.90 | **22.59** | 44.46 | **51.25** | 28.57 | **36.57** | 18.42 | **27.14** | 78.41 | **78.60** | 67.76 | **69.98** | 57.83 | **62.98** |
| Frost | 41.66 | **47.37** | 31.64 | **38.70** | 22.36 | **28.79** | 43.69 | **50.99** | 31.48 | **41.51** | 21.58 | **31.46** | 79.62 | **79.76** | 71.35 | **72.47** | 61.90 | **63.73** |
| Pixelate | 50.48 | **54.57** | 40.79 | **47.37** | 27.51 | **38.02** | 52.46 | **58.02** | 44.19 | **52.26** | 31.50 | **43.28** | 82.29 | 81.39 | 71.55 | **73.95** | 59.78 | **64.11** |
| JPEG | 48.93 | **53.00** | 47.98 | **51.86** | 43.01 | **47.93** | 51.23 | **57.20** | 50.14 | **55.95** | 44.41 | **51.43** | 80.73 | 79.45 | 78.65 | 77.46 | 71.23 | **72.07** |

## 5.2. Robustness Against Common Corruptions

We now analyze the core contribution of Dem-HEC: its ability to enhance model resilience against common corruptions. As shown in Figure 2, high-severity noise destroys image features; therefore, robustness to such severe environmental corruption can reflect the genuine strength of the proposed approach. The jump of up to 54% (RepVGG a0 on CIFAR10) demonstrates that the proposed approach can achieve this.

### 5.2.1. ROBUSTNESS ON CIFAR10-C AND CIFAR100-C

As shown in Table 1 and Table 8 (appendix), applying Dem-HEC leads to dramatic improvements in corruption accuracy (CAcc) across all four architectures tested on CIFAR10-C and CIFAR100-C. The most significant gains on CIFAR10-C are observed for high-frequency noise corruptions. For example, the accuracy of RepVGG-A0 under Gaussian noise at the highest severity (S5) improves from a near-failure rate of 14.37% to 69.29%, which is a relative increase of over 380%. Similarly, under Shot noise, its accuracy improves from 19.30% to 71.40%. This trend is scalable, handling a large number of classes in CIFAR100-C, where RepVGG-A2's accuracy on Shot noise at severity 5 is more than tripled, from 9.49% to 29.28%. A key trend is that the efficacy of Dem-HEC becomes more pronounced as the corruption severity increases. While the baseline models often suffer a catastrophic performance collapse at severity levels 3 and 5, the Dem-HEC-finetuned models exhibit remarkable resilience. For instance, on CIFAR-100-C, the ResNet-20 improves its accuracy against JPEG compression artifacts at S5 from 33.90% to 52.65%. Even for corruption types with relatively strong baselines, such as Snow, Dem-HEC consistently provides a performance boost, increasing ResNet-56 accuracy from 74.35% to 78.39% at S5 on CIFAR10-C. This consistent improvement across diverse models and corruption types validates our hypothesis that encouraging high-entropy, distributed feature representations is a generalizable defense against corruption-induced performance degradation.

### 5.2.2. SCALABILITY AND PERFORMANCE ON LARGE SCALE AND HIGH RESOLUTION DATASETS

The experiment, detailed in Table 2, tests the scalability of Dem-HEC on both CNN and Transformer architectures on Tiny-ImageNet200-C, which features 200 classes and higher-resolution images. For ResNet-18 and ResNet-50, Dem-HEC continues to provide significant robustness gains, boosting ResNet-50's accuracy on Frost corruption at severity 5 (S5) from 21.58% to 31.46%. The analysis of the ViT model, an inherently more robust architecture, offers nuanced insights. While the performance gains from Dem-HEC are more modest than those from CNNs, our method still enhances its resilience, particularly at high corruption severities, such as Snow (improving from 57.83% to 62.98% at S5). The smaller margin suggests that ViT's self-attention mechanism may already promote a "democratic" feature representation. Nevertheless, the ability of Dem-HEC to further improve such a strong baseline underscores its value as a versatile, robustness-enhancing tool. We have also evaluated Dem-HEC on ImageNette and ImageNet-1k; the results are shown in appendix A.2.

Our experiments reported in Table 10 on full ImageNet with merely a 5% class-balanced subset used for finetuning demonstrate that the proposed Dem-HEC with a ViT backbone preserves accuracy on clean images (with a slight drop of 0.04%) but increases robustness across various corruptions by 1-2% on ImageNet-C. It is interesting to note that, despite having minimal training data for each class, the proposed approach demonstrates improved performance across the network and maintains clean accuracy.

*Table 3.* Performance of ViT on ImageNet-1K using 5% images per class, showing average accuracy over corruption severities 1 to 5 before and after applying the proposed Dem-HEC method.

| Setting | Clean | Gaussian | Shot | Impulse | Snow | Frost | Pixelate | JPEG |
|---|---|---|---|---|---|---|---|---|
| Before | 81.09 | 56.46 | 53.74 | 54.28 | 40.81 | 41.18 | 66.30 | 65.76 |
| Dem-HEC | 81.05 | **57.23** | **54.68** | **55.01** | **41.75** | **42.68** | **67.15** | **67.08** |

### 5.2.3. SCALABILITY OF PROPOSED DEM-HEC TO LARGE MODEL

Several studies show that pre-trained models are not robust to corruptions and adversarial perturbations, *including SAM,*

*Table 4.* Robustness evaluation of DINOv3 on CIFAR100 using the proposed Dem-HEC method. The results report average performance across severity levels 1-5 for various corruption types, under both attack and defence settings.

| Setting | Gaussian | Shot | Snow | Frost | Pixelate | JPEG |
|---|---|---|---|---|---|---|
| Attacked | 26.85 | 33.48 | 51.68 | 52.45 | 59.95 | 49.56 |
| Dem-HEC | **29.84** | **36.19** | **54.31** | **54.69** | **61.78** | **52.29** |

*Table 5.* Comparison of corruption-wise accuracy across various robustness methods on Tiny ImageNet dataset with ViT backbone. Our Dem-HEC approach shows the greatest improvement across all corruption categories.

| Corruption | Pad-Crop | TA | BA | BA (AA) | AugMix | AugMax | IPMix | DV+AP+JSD | Dem-HEC |
|---|---|---|---|---|---|---|---|---|---|
| Noise | 20.94 | 25.27 | 21.18 | 24.62 | 29.15 | 30.18 | 28.80 | 34.44 | **60.10** |
| Blur | 17.27 | 31.23 | 18.01 | 28.36 | 30.77 | 31.72 | 28.04 | 37.04 | **62.88** |
| Weather | 13.41 | 22.88 | 13.54 | 20.29 | 19.94 | 20.65 | 20.79 | 29.17 | **70.71** |
| Digital | 20.99 | 33.19 | 21.53 | 33.07 | 32.26 | 33.49 | 32.34 | 40.24 | **67.53** |

*Eva-CLIP, and their variants.* (Al-Tahan et al., 2024; Cui et al., 2024) shows that LMMs are highly vulnerable to visual adversarial attacks, even when such adversaries are crafted with respect to the visual model alone. (Schiappa et al., 2024) shows that Visual Foundation Models (VFMs) struggle with corruptions. (Wang et al., 2024) demonstrates that SAM's performance generally declines under perturbed images, with varying degrees of vulnerability across different perturbations. (Wang et al., 2023) highlights the base-corruption robustness of popular vision backbones, revealing that corruption robustness does not necessarily scale with model or data size. Furthermore, we have evaluated the zero-shot robustness of CLIP with a ViT backbone. On the clean test set of the CIFAR100 dataset, the model achieves 64.18% accuracy, which drops drastically across corruption types and severities. For example, in the case of widespread common corruptions such as Gaussian noise and compression with medium severity (3), the performance drops to 21.65% and 36.15%, respectively. Even at the lowest severity level, performance drops by at least 20%, indicating that these large models are not inherently robust to corruption. For DINOv3, we adopt the standard supervised fine-tuning setup used for self-supervised ViT backbones. We have used the pretrained model with 21.6M parameters, attached a 100-way linear classifier, and fine-tuned the entire model on CIFAR100 using AdamW, analogous to the linear-probe and fine-tuning protocols on downstream classification tasks (Oquab et al., 2024). We then assess robustness by measuring accuracy on CIFAR100-C over multiple corruption types and severities. Similar to CLIP, DinoV3 is also found to be sensitive to corruption, with accuracy dropping to 11% and 34% at compression severity 3 and Gaussian Noise severity 3, respectively. Our experiments also demonstrate a 22% drop in the EVA-CLIP-8B model, proving that these models are not inherently robust. However, our results in Table 4 demonstrate the scalability of our proposed Dem-HEC in defending recent large models[1].

---

[1]The limited boost can also be attributed to limited training due to lack of heavy computing resources

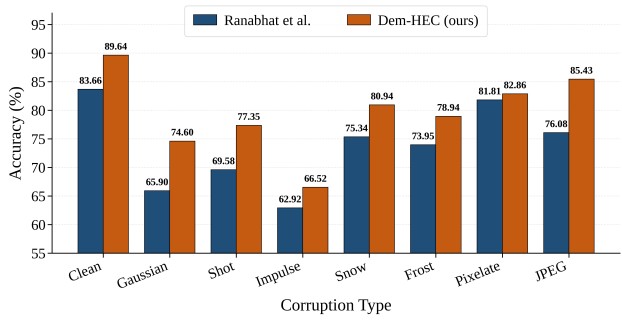

*Figure 6.* Robustness comparison across common corruption types, with results averaged over severity levels 1 to 5. Our method demonstrates consistent improvements over (Ranabhat et al., 2025) across all corruption categories on the CIFAR10 dataset using the ResNet-20 backbone.

### 5.3. Comparison with Existing Robustness Methods

Table 5 presents a systematic comparison of robustness performance across four high-level corruption families: Noise (Gaussian, Shot, Impulse), Blur (Motion, Glass, Defocus, Zoom), Weather (Snow, Frost, Fog, Brightness), and Digital (JPEG compression, Elastic, Pixelate, Contrast). The reported accuracy values represent the mean performance averaged over all individual corruptions and severity levels within each category. We compare Dem-HEC with widely used robustness-enhancing methods, including Pad-Crop, TA (Müller & Hutter, 2021), BA (Hoffer et al., 2020), AA (Cubuk et al., 2019), AugMix (Hendrycks et al., 2020), AugMax (Wang et al., 2021), IPMix (Huang et al., 2023), and DV+AP+JSD (Kim et al., 2025). On Tiny-ImageNet-C, as shown in Table 5, Dem-HEC exhibits substantial improvements, particularly for high-severity corruptions, underscoring its scalability to large-resolution datasets and transformer backbones.

We further included the results against the recent work of (Ranabhat et al., 2025), which proposes a multi-scale push–pull mechanism with channel attention for corruption robustness. As shown in Figure 6, Dem-HEC consistently outperforms this method across all seven corruptions and also improves clean accuracy, indicating that our entropy-guided learning provides stronger feature stability and, as asserted, that the proposed high-entropy samples help reduce the gap between the training and test distributions.

We also compare Dem-HEC with PRIME (Modas et al., 2022) and DiffAug (Sastry et al., 2024) on CIFAR-10-C using WideResNet-18-2. PRIME uses handcrafted augmentation primitives, DiffAug uses diffusion-based synthetic samples, and Dem-HEC learns from bounded high-entropy samples in representation space. Table 6 shows that PRIME gives the strongest standalone corruption accuracy (83.58%), while Dem-HEC clearly improves over DiffAug in both clean accuracy (84.14% → 87.76%) and average corruption accuracy (74.27% → 79.23%). Dem-HEC

*Table 6.* Comparison with PRIME (Modas et al., 2022) and DiffAug (Sastry et al., 2024) on CIFAR-10-C using WideResNet-18-2. We report clean accuracy, corruption-wise accuracy, and the average over all 15 corruption types. Best results are shown in bold.

| Method | Clean | Gaussian | Shot | Impulse | Defocus | Glass | Motion | Zoom | Snow | Frost | Fog | Brightness | Contrast | Elastic | Pixelate | JPEG | Average |
|---|---|---|---|---|---|---|---|---|---|---|---|---|---|---|---|---|---|
| WideResNet | 81.20 | 76.84 | 77.53 | 76.57 | 68.75 | 67.14 | 59.84 | 64.12 | 72.95 | 70.08 | 64.34 | 78.08 | 48.30 | 68.94 | 69.71 | 76.19 | 69.29 |
| Dem-HEC | 87.76 | 82.49 | 83.67 | 80.39 | 81.50 | 77.11 | 75.35 | 80.09 | 80.22 | 78.81 | 74.85 | 84.56 | 61.11 | 81.05 | 82.33 | 84.94 | 79.23 |
| PRIME | 88.42 | 83.82 | 85.09 | 82.10 | **85.88** | **79.31** | **83.63** | **85.37** | 82.30 | 84.50 | **80.94** | 87.77 | **81.71** | 83.39 | 83.42 | 84.48 | **83.58** |
| PRIME + Dem-HEC | **91.41** | 80.83 | 83.45 | 76.92 | 84.88 | 74.56 | 80.71 | 84.26 | **84.18** | **86.02** | 75.33 | **90.36** | 64.20 | **84.26** | **87.54** | **87.50** | 81.67 |
| DiffAug | 84.14 | 79.85 | 80.62 | 79.35 | 76.92 | 71.24 | 66.13 | 73.90 | 75.36 | 74.77 | 69.99 | 81.62 | 51.59 | 75.41 | 76.97 | 80.37 | 74.27 |
| DiffAug + Dem-HEC | 89.33 | **84.74** | **85.81** | **82.87** | 82.56 | 77.85 | 75.51 | 80.60 | 82.03 | 81.73 | 72.96 | 87.15 | 56.66 | 82.07 | 84.40 | 86.75 | 80.25 |

*Table 7.* Loss-term ablation. Avg. denotes the mean over Gaussian, Shot, and Impulse.

| Variant | Clean | Gaussian | Shot | Impulse | Avg. |
|---|---|---|---|---|---|
| **CIFAR100 / RepVGG-A2** | | | | | |
| Dem-HEC | 73.08 | 47.13 | 52.97 | 40.63 | 46.91 |
| w/o InfoNCE | **75.10** | 42.00 | 49.63 | 39.27 | 43.63 |
| w/o KD | 70.90 | **50.90** | **55.83** | **41.17** | **49.30** |
| **ImageNet200 / ResNet50** | | | | | |
| Dem-HEC | 67.82 | 33.32 | 37.58 | 31.35 | 34.08 |
| w/o InfoNCE | **68.21** | 31.05 | 36.32 | 30.86 | 32.74 |
| w/o KD | 63.71 | **34.30** | **38.28** | **31.39** | **34.66** |

also complements both methods: DiffAug + Dem-HEC improves all 15 corruptions and raises clean and average corruption accuracy by 5.19 and 5.98 points, while PRIME + Dem-HEC reaches the best clean accuracy (91.41%).

## 6. Ablation Studies

In this section, we analyze the role of each loss component and the sensitivity of Dem-HEC to its hyperparameters.

**Ablation of loss terms.** Dem-HEC combines clean supervision, high-entropy supervision, InfoNCE, and KD. Table 7 shows that removing *InfoNCE* slightly improves clean accuracy but consistently reduces robustness, confirming that contrastive alignment is the main contributor to corruption-robust representations. For example, on CIFAR100 / RepVGG-A2, the average over Gaussian, Shot, and Impulse drops from 46.91% to 43.63%, and on ImageNet200 / ResNet50 it drops from 34.08% to 32.74%. In contrast, removing *KD* improves the average on these three corruptions but substantially hurts clean accuracy ($73.08\% \rightarrow 70.90\%$ on CIFAR100 and $67.82\% \rightarrow 63.71\%$ on ImageNet200), showing that KD mainly stabilizes the clean-data decision boundary. This trend is consistent with our additional observations on CIFAR10 / ResNet20: without contrastive learning, Gaussian robustness drops from 77.91% to 68.78%, whereas removing KD causes a 7.4-point drop in clean accuracy. Overall, Dem-HEC yields the best balance between clean accuracy and robustness.

**Ablation of hyperparameters.** We study Dem-HEC on CIFAR-10-C with WideResNet-18-2 for 10 epochs. The method is relatively stable with respect to $\alpha$ and $\lambda_C$, while $\lambda_{KD}$ is the most sensitive parameter. For example, increas-

ing $\alpha$ from 0.0 to 0.5 steadily improves clean accuracy from 84.00% to 84.89% and average corruption accuracy from 78.21% to 79.19%. In contrast, $\lambda_C$ shows only mild sensitivity: small-to-moderate values work well, while very large values slightly reduce both clean and corruption accuracy. The strongest sensitivity appears in $\lambda_{KD}$, where increasing it from 0.0 to 2.0 reduces clean accuracy from 89.30% to 77.88% and average corruption accuracy from 82.79% to 71.83%, indicating that overly strong distillation over-constrains adaptation.

We also analyze the high-entropy sample generator. Increasing the perturbation radius $\epsilon$ from $1/255$ to $16/255$ improves average corruption accuracy from 78.79% to 80.48%, especially at S5 ($75.04\% \rightarrow 79.33\%$), but an excessively large radius ($32/255$) degrades both clean accuracy (77.67%) and corruption accuracy (76.24%). For the number of PGA steps $T_{he}$, we do not observe instability: increasing $T_{he}$ from 1 to 20 changes average corruption accuracy only marginally ($78.68\% \rightarrow 78.80\%$), but raises runtime from 178.9s to 955.3s. Thus, moderate $\epsilon$ and small-to-moderate $T_{he}$ provide the best practical trade-off. In the end, note that the main computational overhead of Dem-HEC over standard fine-tuning comes from high-entropy sample generation and the extra frozen-teacher forward pass.

## 7. Conclusion

The work extensively highlights the vulnerability of shallow to large foundation models, including CLIP and DinoV3. It raises an urgent demand for a cost-effective plug-in that can ensure the robustness of these already developed models against natural corruptions. To address this, we introduced Dem-HEC, a novel fine-tuning framework that enhances the robustness of pre-trained models through synthetically generated high-entropy samples. By combining contrastive representation alignment with dual cross-entropy and knowledge distillation, our method learns to produce stable predictions even when internal features are maximally uncertain. Extensive evaluations across CIFAR10, CIFAR100, and Tiny-ImageNet demonstrate that Dem-HEC significantly enhances robustness against a wide array of corruptions and severities without compromising performance on clean data. The framework's effectiveness across diverse architectures, including both CNNs and Vision Transformers, validates our approach as a scalable and generalizable solution.

## Acknowledgements

V. Kumar is partially supported through the Visvesvaraya PhD Fellowship. We also thank the reviewers for the valuable comments and suggestions, which helped improve this paper.

## Impact Statement

This paper presents the first-ever study to inherently make the deep learning models robust against natural image corruptions. The work will ensure the deployment of the current deep models for the real benefit of society by reducing their limitations in handling the "true" and "actual" distribution of the real world.

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

*Table 8.* Corruption Accuracy (CAcc.) on CIFAR100-C, comparing performance before and after applying Dem-HEC. Our method yields significant robustness gains across all models, particularly for noise-based corruptions and at higher severity levels (S3 and S5).

| Backbone | ResNet20 | | | | | | ResNet56 | | | | | |
|---|---|---|---|---|---|---|---|---|---|---|---|---|
| Severity | S1 | | S3 | | S5 | | S1 | | S3 | | S5 | |
| Corruption | Before | After | Before | After | Before | After | Before | After | Before | After | Before | After |
| Gaussian | 33.18 | **59.17** | 11.76 | **32.13** | 8.26 | **19.78** | 37.06 | **64.47** | 13.13 | **34.46** | 8.11 | **22.97** |
| Shot | 43.97 | **62.23** | 15.93 | **40.91** | 9.09 | **21.84** | 49.32 | **67.11** | 18.18 | **43.72** | 9.62 | **25.31** |
| Impulse | 46.87 | **56.13** | 17.98 | **29.87** | 4.82 | **6.23** | 49.50 | **59.22** | 18.19 | **31.15** | 4.68 | **8.65** |
| Snow | 57.44 | **58.88** | 45.63 | **49.45** | 36.92 | **43.83** | 62.21 | **64.73** | 50.57 | **55.53** | 40.96 | **48.66** |
| Frost | 56.38 | **61.29** | 36.16 | **47.24** | 25.51 | **38.09** | 60.06 | **65.88** | 39.75 | **51.60** | 29.25 | **42.63** |
| Pixelate | 63.08 | 62.55 | 43.76 | **57.69** | 13.65 | **36.46** | 66.41 | **68.02** | 50.11 | **64.25** | 18.41 | **47.27** |
| JPEG | 51.48 | **60.49** | 40.74 | **56.31** | 33.90 | **52.65** | 53.36 | **64.43** | 42.89 | **60.25** | 35.21 | **57.05** |

| Backbone | RepVGG_a0 | | | | | | RepVGG_a2 | | | | | |
|---|---|---|---|---|---|---|---|---|---|---|---|---|
| Severity | S1 | | S3 | | S5 | | S1 | | S3 | | S5 | |
| Corruption | Before | After | Before | After | Before | After | Before | After | Before | After | Before | After |
| Gaussian | 38.56 | **68.31** | 11.62 | **42.52** | 6.99 | **28.42** | 39.44 | **68.46** | 12.76 | **40.54** | 7.83 | **27.61** |
| Shot | 51.35 | **70.67** | 17.45 | **50.86** | 8.30 | **30.92** | 54.18 | **71.05** | 19.19 | **49.61** | 9.49 | **29.28** |
| Impulse | 54.65 | **65.46** | 21.74 | **43.46** | 6.50 | **15.22** | 55.89 | **65.27** | 22.22 | **41.20** | 7.13 | **13.78** |
| Snow | 66.86 | **68.38** | 55.98 | **58.51** | 46.93 | **52.52** | 68.22 | **69.12** | 57.84 | **59.77** | 47.55 | **53.11** |
| Frost | 65.95 | **69.53** | 46.14 | **56.63** | 35.01 | **48.31** | 67.30 | **70.41** | 47.22 | **57.97** | 36.40 | **49.60** |
| Pixelate | 70.29 | **71.02** | 56.98 | **68.01** | 21.27 | **52.02** | 73.02 | **71.87** | 59.46 | **69.39** | 23.11 | **55.79** |
| JPEG | 60.58 | **68.09** | 50.55 | **64.76** | 43.81 | **61.38** | 62.55 | **69.28** | 52.31 | **65.89** | 44.88 | **62.98** |

# A. Appendix

## A.1. Common Corruption

In this work, we focus on seven widely recognized corruption types that reflect real-world degradations encountered in image acquisition, transmission, and storage. The first category consists of additive noise corruptions: *Gaussian noise*, which is a common disturbance in low-light conditions or faulty sensor environments, modelled as a signal-independent additive noise with a zero-mean Gaussian distribution. *Shot noise*, also referred to as Poisson noise, arises from the discrete nature of photons in optical sensors and is particularly prevalent in low-exposure or high-sensitivity imaging scenarios. *Impulse noise*, the color analogue of salt-and-pepper noise, appears due to bit errors in transmission or malfunctioning pixels in digital sensors, introducing sharp intensity spikes. The second category involves environmental corruptions. *Snow corruption* introduces white, irregular particles across the scene, imitating obstructive precipitation that reduces visibility and alters texture distribution. *Frost corruption* mimics the accumulation of ice crystals on a lens or window surface, producing distortions similar to imaging through frozen glass. Both snow and frost alter the global scene's appearance and occlude local details, challenging a model's ability to extract meaningful representations. Finally, we consider digital corruption, a consequence of post-capture transformations. *Pixelation* occurs when low-resolution images are upsampled, leading to blocky structures and loss of fine details, a phenomenon frequently observed in digital zoom or low-bandwidth video transmission. *JPEG compression* is a widely used lossy encoding scheme for digital storage and web transmission, where aggressive compression at high ratios introduces block artifacts and loss of high-frequency details. Together, these seven corruption types cover a broad range of sensor-level, environmental, and digital artifacts, providing a comprehensive testbed for evaluating the robustness of deep neural networks against corruption. Moreover, for comprehensiveness, each corruption has been applied with multiple severities reflecting mild (S1), medium (S3), and high (S5) severity. The corresponding severity parameter has been inspired by the work of (Hendrycks & Dietterich, 2019) and is given at[2]. Figure 2 shows the challenge that the proposed research is handling by tackling the loss of visual cues at high severities, and the strength of the proposed research.

---

[2]https://github.com/bethgelab/imagecorruptions

## A.2. Robustness Scalability on High Resolution Dataset

We evaluated Dem-HEC on ImageNette, a 10-class subset of the original ImageNet dataset, using AlexNet. Since the ImageNette image resolution is aligned with that of full ImageNet, the consistently improved performance of Dem-HEC demonstrates its image-resolution-agnostic nature in handling image corruption. Robustness of the proposed Dem-HEC on the ImageNet subset with the AlexNet backbone is shown in Table 9. Please note that the proposed approach not only increases corruption robustness but also retains clean accuracy, clearly demonstrating the ideal trade-off between accuracy and robustness. Our experiments on full ImageNet with a 5% class-balanced subset used for finetuning demonstrate that the proposed Dem-HEC with a ViT backbone preserves accuracy on clean images (with a slight drop of 0.04%) but increases robustness across various corruptions by 1-2% on ImageNet-C. It is interesting to note that, despite having minimal training data for each class, the proposed approach demonstrates improved performance across the network and maintains clean accuracy.

*Table 9.* Performance before and after applying the proposed Dem-HEC method across common corruption types on the ImageNet subset using the AlexNet backbone, showing average accuracy over corruption severities 1 to 5.

| Method | Clean | Gaussian | Shot | Impulse | Snow | Frost | Pixelate | JPEG |
|---|---|---|---|---|---|---|---|---|
| Before | 91.0 | 53.89 | 52.30 | 49.46 | 63.77 | **70.57** | 73.41 | 85.21 |
| After (Dem-HEC) | 91.01 | **56.29** | **54.54** | **51.37** | **67.26** | 67.18 | **80.16** | **88.03** |

*Table 10.* Performance of ViT on ImageNet-1K using 5% images per class, showing average accuracy over corruption severities 1 to 5 before and after applying the proposed Dem-HEC method.

| Setting | Clean | Gaussian | Shot | Impulse | Snow | Frost | Pixelate | JPEG |
|---|---|---|---|---|---|---|---|---|
| Before | 81.09 | 56.46 | 53.74 | 54.28 | 40.81 | 41.18 | 66.30 | 65.76 |
| After (Dem-HEC) | **81.05** | **57.23** | **54.68** | **55.01** | **41.75** | **42.68** | **67.15** | **67.08** |

## A.3. Robustness Against Complex or Combined Corruption

To further validate the effectiveness of the proposed approach, we conducted a study by combining multiple corruptions (applied together sequentially), such as Gaussian noise and shot noise. As shown in Table 11, the proposed Dem-HEC increases performance on the combined (complex) corruption by 30% when both noises are applied with severity 1. Even at the higher severity level(S2), the performance increases by approximately 8%. These experiments are conducted on the CIFAR-10 dataset using the ResNet-20 architecture.

*Table 11.* Robustness results of ResNet20 on CIFAR10 combining Gaussian and Shot noise corruptions across severity levels (S1 to S5).

| Method | S1 | S2 | S3 | S4 | S5 |
|---|---|---|---|---|---|
| Before | 17.42 | 13.32 | 12.18 | 11.31 | 10.75 |
| After (Dem-HEC) | **47.87** | **21.29** | **13.63** | **12.08** | **12.14** |

We have evaluated the performance of the proposed approach against various blur corruptions. Table 12 here refers to the increased performance of the proposed robust model compared to the pre-trained (or non-robust model). The improvement across such different forms of corruption suggests that the proposed algorithm can handle vast corruption groups effectively and surpass existing state-of-the-art works by a significant margin.

## A.4. Implementation Details

Our proposed framework, Dem-HEC, is implemented in PyTorch. For CIFAR10 and CIFAR100, we adopt four pretrained architectures, variants of ResNet and RepVGG from (Chen). Apart from this, all other models are directly taken from the PyTorch library, using ImageNet-pretrained weights, and then finetuned for up to 5 epochs on each dataset's training set before applying our proposed Dem-HEC approach. We finetune all pretrained models for 20 epochs using a batch size of 128 and the full training sets of CIFAR10, CIFAR100, and Tiny-ImageNet. The training is performed using a Stochastic Gradient Descent (SGD) optimizer with an initial learning rate of 0.05, momentum of 0.9, and weight decay of $5 \times 10^{-4}$. A cosine annealing learning rate schedule with a 2-epoch linear warmup phase is employed. The parameter $\lambda_{KD} = 0.5$, and $T = 2.0$ has been taken for the $L_{KD}$. $\lambda_C = 1.0$ and temperature $\tau = 0.2$ has been taken for $L_{InfoNCE}$. The parameters for the CIFAR10 and CIFAR100 experiments are identical. In contrast, the Tiny-ImageNet experiment uses a smaller batch

*Table 12.* Performance improvement (%) of ResNet-50 on the Tiny-ImageNet dataset under different blur corruptions and severity levels after applying the proposed Dem-HEC method.

| Severity | Defocus Blur | Glass Blur | Motion Blur | Zoom Blur |
|---|---|---|---|---|
| 1 | 5.47 | 5.04 | 5.71 | 5.85 |
| 3 | 2.99 | 4.69 | 6.08 | 4.77 |
| 5 | 2.30 | 0.96 | 4.92 | 3.61 |

size=32, fewer epochs=10, and a lower learning rate=0.0005 to accommodate the larger Vision Transformer (ViT-L) model and higher-resolution images=$384 \times 384$. All experiments are conducted on a machine with a 104-Core 2.0GHz CPU, 251GB of system memory, and an NVIDIA RTX A6000 GPU with 47 GB of memory.

