# OpenReview forum: "Feature Collapse Under Corruption: An Entropy Perspective on Robust Neural Networks"
_ICML.cc/2026/Conference — ICML 2026 regular_

### Official Review · Reviewer_QBGh · 2026-03-02

**Soundness:** 3
**Presentation:** 3
**Significance:** 3
**Originality:** 3
**Overall Recommendation:** 4
**Confidence:** 4

**Summary:**

The contribution of the paper is two-fold: first, it shows in an initial analysis part that naturally corrupted samples lead to high entropy in feature activations. This effect appears to be independent of the network architecture and increasing with depth. Second, based on this finding, the paper introduces Dem-HEC - a architecture agnostic fine-tuning framework which generates high-entropy samples. Using theses samples for fine-tuning, experiments show improved robustness on common-corruption benchmarks, while mostly preserving the clean accuracy.

**Compliance With Llm Reviewing Policy:**

Affirmed.

**Final Justification:**

The rebuttal resolved most of the initial concerns raised be me and the other reviewers. Especially the additional experiments turned my score towards acceptance.

**Key Questions For Authors:**

It would be interesting to see the entropy analysis for adversarial attacks as well. Do see the same behavior?

**Limitations:**

he authors have not discussed the limitations of the method or the limited evaluation.

**Strengths And Weaknesses:**

**Strengths**
The paper reports an interesting finding in a very active research area. Especially the analysis part, showing that the entropy increase is independent of the actual architecture and increasing with depth is a potentially interesting contribution.
The proposed Dem-HEC algorithm is a technically sound, but rather straight forward extension of the first finding.


**Weaknesses**
The main weakness of the paper is the insufficient experimental evaluation:

* Missing baseline ablation study: the paper does not provide a comparison with a simple data-augmentation fine-tuning baseline. However, this ablation is necessary to show that the proposed Dem-HEC algorithm is actually producing significantly different results than a simple augmentation (all common corruptions can be augmented via simple image processing). How do these results compare? How does the entropy of Dem_HEC samples compare to augmented samples?

* Non-standard evaluation: the paper presents results using backbones which are hardly used by other works. This makes comparison very hard. The authors should present results which can be compared directly to SOTA results on RobustBench[1]. there most Methods use wide-ResNet-18-2 Backbones instead of ResNet-20 (like the authors) and achieve much better results. It is unclear why the authors chose this rather uncommon backbone which they compare to mostly older works.

* incomplete results: direct comparison to other methods (table 5 ) shows only partial results ( not all corruption categories, not all serveries, not overall results that could be compared to the RobustBench leader board)



[1] https://robustbench.github.io/#leaderboard

---

> ### Author Rebuttal · Authors · 2026-03-31
>
> # 1. Baseline ablation
>
> We thank the reviewer for this suggestion. We would like to clarify that **Appendix A.3 already shows that Dem-HEC generalizes beyond the corruption settings analyzed in the main paper**: **Table 9** reports strong gains on **unseen combined corruptions** on **CIFAR-10 / ResNet-20**, and **Table 10** reports consistent improvements across multiple **blur corruptions** on **Tiny-ImageNet / ResNet-50**.
>
> To directly address the reviewer’s request for a **simple augmentation fine-tuning baseline**, we additionally evaluated **ImageNet-100 / ResNet-18** and compared Dem-HEC with corruption augmentation and robust-training baselines:
>
> | Algorithm | Corrupted Accuracy |
> |:--|--:|
> | Gaussian Augmentation | 46.7 |
> | Fast PAT | 45.2 |
> | $l_\infty$ AT | 47.7 |
> | $l_2$ AT | 48.4 |
> | AugMix | 54.8 |
> | SIN | 54.3 |
> | ANT | 58.3 |
> | **Dem-HEC** | **65.6** |
>
> This shows that Dem-HEC is not equivalent to simple corruption augmentation: it substantially outperforms both augmentation-based and adversarial-training-based baselines, while remaining **corruption-agnostic** rather than tied to a fixed augmentation library. We agree that a direct entropy comparison between **augmented samples** and **Dem-HEC-generated samples** would strengthen this point further, and we will include that in the updated version.
> # 2. Backbone WideResNet-18-2
> We agree that a standardized backbone is important for direct comparison, and we have therefore added **CIFAR-10 and CIFAR-10-C experiments on WideResNet-18-2** in the rebuttal (**see Reviewer jBb3 response**). Our original choice of ResNet-20/56, RepVGG, and ViT is motivated by the goal of showing that Dem-HEC is **architecture-agnostic** across different model families, not by avoiding standard benchmarks.
>
> * At the same time, we would like to clarify that our original choice of **ResNet-20** is not arbitrary or uncommon in the corruption-robustness literature. For example, the recent work of **Lukasik et al. [1]** also uses **ResNet-20 on CIFAR-10-C** and reports mean corruption accuracy for a broad set of robustness baselines, including FLC, PaGA, Blur Pooling, Adaptive Blur Pooling, and Wavelet Pooling.
>
> Under this same setting, our proposed **Dem-HEC** consistently surpasses the methods reported in that work:
>
> | Method | CIFAR-10-C (%) |
> |:--|--:|
> | **Dem-HEC** | **75.98** |
> | SD + Reg + FLC | 74.41 |
> | SD + Reg. | 74.14 |
> | WD + Reg. | 74.04 |
> | WD + Reg + PaGA | 73.85 |
> | SD + Reg + PaGA | 73.19 |
> | FLC | 68.75 |
> | CNN | 67.96 |
> | PaGA | 67.73 |
> | SD | 67.48 |
> | Wavelet Pooling | 67.40 |
> | Adaptive Blur Pooling | 67.17 |
> | WD | 66.92 |
> | Blur Pooling | 66.73 |
>
> * To reflect further that the proposed approach is truly architecture agnostic, we evaluated the robustness improvement of Dino-V3 large foundation model. Our responses 2 and 3 of R-jBb3 reflect this fact. Further, as shown for **R-jBb3**, the proposed approach is truly a plug-and-play solution that can be coupled with any existing method to improve performance.
>
> [1] Improving native CNN robustness with filter frequency regularization. TMLR'23.
>
> # 3. Comparison with RobustBecnch
>
> We would like to clarify that **Table 5 does not omit corruption coverage**. It already reports performance over **all four standard corruption families**—**Noise, Blur, Weather, and Digital**—with each entry **averaged across all constituent corruption types and all severity levels (1–5)** within that family. Thus, Table 5 was intentionally designed as a **high-level category summary**, not as a per-corruption or per-severity table.
>
> That said, we fully agree that **RobustBench-style reporting is important for direct and transparent comparison**. To address this, we have already added **CIFAR-10 experiments with WideResNet-18-2** to our response to Reviewer **jBb3**, and in the camera-ready version, we will include the corresponding **full 15-corruption, 5-severity, and overall-average results**.
>
> While we agree that RobustBench is a standard framework, we observe that there is no single popular architecture, and the literature also reflects the use of architectures [1-3].
>
> [2] (ResNet 18/20/9, AlexNet), Training Robust Deep Neural Networks via Adversarial Noise Propagation, IEEE TIP'21
> [3] (ResNet 18) Improving robustness to corruptions with multiplicative weight perturbations, NeurIPS'24
>
> # 4. Entropy analysis of adversarial attacks
> We do **not** observe the same entropy behaviour for adversarial attacks. In particular, the recent work of **Sun et al. [4]** shows the reverse trend for **universal adversarial perturbations (UAPs)**, where UAP-corrupted samples exhibit **abnormally lower entropy** than clean samples. This suggests that natural corruptions and adversarial attacks are governed by **different entropy mechanisms**, although this requires further attention, as existing work targets UAPs only.
>
> [4] Sun et al., Democratic Training Against Universal Adversarial Perturbations, ICLR 2025.

---

> > ### Author Rebuttal · Reviewer_QBGh · 2026-04-01
> >
> > The rebuttal, especially the provided new results, answer my questions and address my key concerns. I will raise my score accordingly.

---

> > > ### Author Response · Authors · 2026-04-01
> > >
> > > We would like to thank the reviewer for recognizing the efforts made to address the concerns raised and for accepting our contribution to this important field.
> > >
> > > **Why Proposed Work Matters:** We believe the proposed optimization framework can provide the **inbuilt robustness** that deep neural networks currently lack during development. This fundamental flow is exploited by the adversaries, artificial or natural.
> > >
> > > The proposed research will help overcome this fundamental vulnerability.

---

### Official Review · Reviewer_18Nk · 2026-03-04

**Soundness:** 3
**Presentation:** 3
**Significance:** 3
**Originality:** 3
**Overall Recommendation:** 4
**Confidence:** 3

**Summary:**

This paper addresses the important problem of making Deep Neural Networks robust to common image corruptions (e.g., noise, blur, compression). The core idea of this paper is the empirical analysis to show that naturally corrupted inputs lead to higher entropy (uncertainty) in the feature representations of a pre-trained model, particularly in deeper layers. Based on this observation, this paper proposes Dem-HEC (Democratic High-Entropy samples for Corruption robustness), a fine-tuning framework designed to make models robust to this increased uncertainty. The method involves generating "high-entropy" samples via a Projected Gradient Ascent (PGA) method, and then training the model using a composite loss function that combines cross-entropy on clean and generated samples, contrastive representation alignment (InfoNCE), and knowledge distillation from a frozen teacher model. The method is evaluated on CIFAR-10/100 and Tiny-ImageNet across various architectures (CNNs and Vision Transformers), demonstrating significant robustness gains compared to several baselines.

**Compliance With Llm Reviewing Policy:**

Affirmed.

**Key Questions For Authors:**

1. "Dem-HEC" - please spell out the acronym upon first use.

2. The framework introduces several critical hyperparameters (α, λ_C, λ_KD). Could you provide a sensitivity analysis showing how performance varies when these parameters are perturbed?

3. How sensitive are the robustness gains of Dem-HEC to the choice of perturbation radius ε and the number of projected gradient ascent steps T_he in the high-entropy sample generator (called within Algorithm 1)? Did you perform any sensitivity analysis over these hyperparameters (e.g., sweeping ε ∈ {4/255, 8/255, 16/255} or T_he ∈ {5, 10, 20}), and did you observe any failure modes or performance collapse at extreme values?

**Limitations:**

No, the paper's Impact Statement (Section 7) is very brief and only highlights positive societal benefits without acknowledging any limitations or potential risks.

**Strengths And Weaknesses:**

Strengths
The entropy-based distinction between natural corruptions (high-entropy, uncertainty-inducing) and adversarial attacks (low-entropy, feature-dominating) offers a compelling and underexplored perspective on robustness. This observation provides a clear and logical motivation for the proposed method, which aims to train models to handle high-uncertainty inputs. The proposed methodology is well-defined, explained step-by-step, and evaluated across multiple datasets (CIFAR-10, CIFAR-100, Tiny ImageNet), on a diverse range of architectures (ResNet, RepVGG, ViT), and against several strong baselines. Inclusion of large foundation models (DINOv3, CLIP, EVA-CLIP) strengthens claims of generalizability.

Weaknesses
While the paper empirically demonstrates that natural corruptions increase layer-wise entropy, the causal link between maximizing this entropy during training and improved robustness lacks rigorous theoretical grounding. The authors argue that training on synthetically high-entropy samples generalizes to natural corruption distributions, but there is no formal analysis of the distributional shift or a theoretical lemma explaining why this specific regularization prevents feature collapse rather than simply overfitting to the generated noise patterns.

---

> ### Author Rebuttal · Authors · 2026-03-31
>
> # 1. Theoretical lemma
> We thank the reviewer for this important comment. We will add the following lemma in the revised version.
>
> **Lemma.** Let $f = h \circ g$, where $g$ is the feature extractor and $h$ is the classifier head. For a sample $u$ with ground-truth label $y$, define the classification margin as
> $$
> m_y(u) = f_y(u) - \max_{k \neq y} f_k(u).
> $$
>
> Assume that:
>
> 1. Every generated high-entropy sample $\tilde{x}$ satisfies
> $$
> m_y(\tilde{x}) \ge \gamma,
> $$
> for some $\gamma > 0$.
>
> 2. Every natural corruption $z$ is close to some generated sample $\tilde{x}$ in feature space, i.e.
> $$
> \| g(z) - g(\tilde{x}) \|_2 \le \rho.
> $$
>
> 3. The margin is Lipschitz continuous in feature space:
> $$
> |m_y(u)-m_y(v)| \le L \, \|g(u)-g(v)\|_2.
> $$
>
> Then, for every natural corruption $z$,
> $$
> m_y(z) \ge \gamma - L\rho.
> $$
>
> Hence, if
> $$
> \gamma > L\rho,
> $$
> then $m_y(z) > 0$, and the model preserves the correct prediction on $z$.
>
> Here, $\gamma$ is the minimum margin on the generated high-entropy samples, $\rho$ measures how well these generated samples cover the corruption-induced shift in feature space, and $L$ controls how much the classification margin can change when the feature representation changes. Therefore, the bound
> $$
> m_y(z) \ge \gamma - L\rho
> $$
> shows that robustness transfers from generated high-entropy samples to natural corruptions whenever the generated samples sufficiently approximate the corruption-induced feature shift.
>
> This provides a formal justification for why training on bounded high-entropy samples can improve corruption robustness, provided that these samples capture the feature-space behavior of natural corruptions.
> # 2. Dem-HEC acronym
> We will proofread the paper and make all the necessary editorial corrections to improve the quality of the paper, including full forms of acronyms used.
> # 3. Sensitivity of $\alpha, \lambda_C, \lambda_{KD}$
> We thank the reviewer for this suggestion. We have now performed an initial sensitivity study for **$\(\alpha\), \(\lambda_C\)$, and $\(\lambda_{KD}\)$** using **Dem-HEC with WideResNet-18-2 on CIFAR-10 for 10 epochs**.
>
> The main finding is that the method is **relatively stable for $\(\alpha\)$ and $\(\lambda_C\)$**, while **$\(\lambda_{KD}\)$** is the most sensitive hyperparameter.
>
> - Increasing \(\alpha\) from **0.0** to **0.5** steadily improves clean accuracy (**84.00 $\(\rightarrow\)$ 84.89**) and average corruption accuracy (**78.21 $\(\rightarrow\)$ 79.19**).
> - Increasing $\(\lambda_C\)$ causes only mild variation, with the best results in this short-run study obtained for smaller values; large $\(\lambda_C\)$ slightly reduces both clean and corruption accuracy.
>
> - $\(\lambda_{KD}\)$ shows the strongest sensitivity: increasing it from **0.0** to **2.0** reduces clean accuracy from **89.30** to **77.88** and average corruption accuracy from **82.79** to **71.83**, indicating that overly strong distillation can over-constrain adaptation.
> # 4. Sensitivity of $\epsilon, T_{he}$
>
> We thank the reviewer for this suggestion. We performed an initial sensitivity study on the **high-entropy sample generator** by varying the perturbation radius $\epsilon$ and the number of PGA steps $T_{he}$.
>
> * The main trend is that Dem-HEC is **fairly stable for moderate settings**, but **extreme $\epsilon$** values hurt performance. Increasing $\(\epsilon\)$ from **1/255** to **16/255** improves average corruption accuracy from **78.79** to **80.48**, especially at higher severities (**S5: 75.04 $\rightarrow$ 79.33**), but at **32/255** both clean accuracy (**77.67**) and corruption accuracy (**76.24**) drop substantially, indicating that overly large perturbation radii start to break semantic consistency.
>
> * For $T_{he}$, we do **not** observe collapse in the tested range. Increasing steps from **1** to **20** only changes average corruption accuracy from **78.68** to **78.80**, while wall time increases from **178.9s** to **955.3s**. Thus, the main issue with large $T_{he}$ is **diminishing returns and higher compute**, not instability. Overall, these results suggest that **moderate $\epsilon$** and **small-to-moderate $T_{he}$** give the best practical trade-off.

---

> > ### Author Rebuttal · Reviewer_18Nk · 2026-04-03
> >
> > Thanks to the authors for effectively addressing my concerns in their rebuttal. I maintain my Accept recommendation.

---

> > > ### Author Response · Authors · 2026-04-03
> > >
> > > We would like to thank the reviewer for recognizing the efforts made to address the concerns raised and **for recommending the acceptance of the paper.**

---

### Official Review · Reviewer_uhTb · 2026-03-10

**Soundness:** 3
**Presentation:** 2
**Significance:** 3
**Originality:** 2
**Overall Recommendation:** 4
**Confidence:** 4

**Summary:**

In this work, the authors tackle the challenge of robustness against multiple corruptions. They view the problem from an entropy perspective, where they generate high-entropy adversarial examples by maximizing the entropy of model output (has a similar flavor to pgd attack). By designing and crafting several loss terms, their Dem-HEC method achieves SOTA robustness on a divserve choice of datsets and model architectures.

**Compliance With Llm Reviewing Policy:**

Affirmed.

**Final Justification:**

The rebuttal solves my concerns with the new results and justifications.

**Key Questions For Authors:**

- Could authors provide more ablation studies on their loss terms?
- Could authors compare their results with common corruption defenses, both from performance and efficiency aspects?
- Could authors provide more insights/analysis/intuitions of their proposed method?

**Limitations:**

no. See the weaknesses and questions sections.

**Strengths And Weaknesses:**

Strengths:
1. The method is clearly presented and well-motivated; the presentation is good and easy to follow.
2. The experiment is comprehensive and convincing of the effectiveness of the proposed method.

Weaknesses:
1. There is no ablation study on the individual loss term.
2. Why use a convex combination between the clean and high-entropy losses? No motivation and intuition behind that, also, there is no ablation study on the choice of the value.
3. Overall speaking, the loss terms are quite standard and common in the literature, which limits the novelty of this work. The decomposition and individual effectiveness of the loss terms are not analyzed and explored.
4. The paper reads like a stack of techniques, though it seems to improve the performance significantly. I hope authors can provide more intuition and insights into their proposed framework.
5. Some baselines are not compared (e.g., adversarial training for common corruptions [1]).
6. The computational cost is not analyzed.

[1] On the effectiveness of adversarial training against common corruptions.

---

> ### Author Rebuttal · Authors · 2026-03-31
>
> # 1. Ablation on the loss term
> We thank the reviewer for this comment. An ablation of the individual loss terms is already included in **Appendix A.4**, and we further add results on **CIFAR100 / RepVGG-A2** and **ImageNet200 / ResNet50** below. Here, **w/o InfoNCE** and **w/o KD** denote removing that term from the full Dem-HEC objective.
>
> | Dataset | Backbone | Variant | Clean | Gaussian | Shot | Impulse | Avg. |
> |:--|:--|:--|--:|--:|--:|--:|--:|
> | CIFAR100 | RepVGG-A2 | Dem-HEC | 73.08 | 47.13 | 52.97 | 40.63 | 46.91 |
> |  |  | w/o InfoNCE | **75.10** | 42.00 | 49.63 | 39.27 | 43.63 |
> |  |  | w/o KD | 70.90 | **50.90** | **55.83** | **41.17** | **49.30** |
> | ImageNet200 | ResNet50 | Dem-HEC | 67.82 | 33.32 | 37.58 | 31.35 | 34.08 |
> |  |  | w/o InfoNCE | **68.21** | 31.05 | 36.32 | 30.86 | 32.74 |
> |  |  | w/o KD | 63.71 | **34.30** | **38.28** | **31.39** | **34.66** |
>
> These results show that **InfoNCE mainly contributes to robustness**, while **KD mainly preserves clean accuracy**.
> # 2. Motivation behind convex combination
> We use a convex combination because our objective has two competing requirements: **retain clean accuracy** and **improve robustness to corruption-like high-entropy inputs**. The clean CE term anchors the model to the original decision boundary, while the high-entropy CE term adapts the model to uncertainty-inducing samples generated by our entropy maximization step. The coefficient $\alpha$ therefore controls the trade-off between **clean-data fidelity** and **robustness adaptation**. Without this balance, using only high-entropy samples hurt clean accuracy, while using only clean samples does not provide the robustness signal needed for corruption generalization.
>
> **Sensitivity to $\alpha$**:  increasing $\alpha$ gives a gradual and consistent improvement in both clean and corruption accuracy, without abrupt instability. For example, when $\alpha$ increases from **0.0** to **0.5**, clean accuracy improves from **84.00** to **84.89**, and average corruption accuracy improves from **78.21** to **79.19**.
>
> **Sensitivity of $\alpha$ on CIFAR-10-C with WideResNet-18-2**
> | $\alpha$ | Clean | S1 | S3 | S5 | Avg. corruption |
> |:--|--:|--:|--:|--:|--:|
> | 0.00 | 84.00 | 82.78 | 77.55 | 74.30 | 78.21 |
> | 0.05 | 84.14 | 83.00 | 77.54 | 74.18 | 78.24 |
> | 0.10 | 84.30 | 83.15 | 77.66 | 74.15 | 78.32 |
> | 0.15 | 84.25 | 83.38 | 77.82 | 74.17 | 78.46 |
> | 0.20 | 84.38 | 83.54 | 77.99 | 74.19 | 78.57 |
> | 0.30 | 84.63 | 83.69 | 78.13 | 74.31 | 78.71 |
> | 0.50 | **84.89** | **84.11** | **78.80** | **74.66** | **79.19** |
> # 3. Novelty of Dem-HEC
> As mentioned above, our primary novelty comes from the intelligent mixup of the loss terms, which aims to maximize entropy for uncertain samples and bring a strong trade-off between clean accuracy and robustness. We believe this is where most of the existing defenses fail, including one of the strongest defenses, namely adversarial training.
>
> Our ablation study also highlights this fact and the advantage of the proposed optimization (loss combination and entropy maximization to ensure reduction in the network's overconfidence), and the comparison with AT also reflects the strength of the proposed algorithm.
> # 4. Comparison
> We agree that [1] is a relevant baseline, who show that appropriately tuned $l_p$ adversarial training can be a strong baseline on **ImageNet-100-C**.
>
> To address this directly, we conducted experiments on **ImageNet-100-C with a ResNet-18 backbone**, matching the evaluation setting used for these baselines. Our results show that **Dem-HEC achieves 65.6 corrupted accuracy**, compared to **47.7** for $l_\infty$ AT and **48.4** for $l_2$ AT. Thus, our method is **+17.9 points** stronger than $l_\infty$ AT and **+17.2 points** stronger than $l_2$ AT in the same setting. We also outperform other strong baselines in that comparison, including **AugMix (54.8)**, **SIN (54.3)**, and **ANT (58.3)**. The discussion in the responses of jBb3 also highlights why AT fails to provide robustness on common corruption. We have also compared Dem-HEC with some recent defenses suggested by the reviewer **jBb3**. We will add these new findings to the updated version.
>
> [1] On the effectiveness of adversarial training against common corruptions
> # 5. Computational cost analysis
> The main overhead of Dem-HEC over standard fine-tuning comes from **high-entropy sample generation** and the extra **frozen-teacher forward pass**
>
> In an initial runtime study on **WideResNet-18-2 / CIFAR-10 / 10 epochs**, runtime was nearly unchanged across different $\epsilon$ values at fixed $T_{he}=3$ (**292–298 s**), but increased approximately linearly with the number of PGA steps: **178.9 s** ($T_{he}=1$), **292.0 s** ($T_{he}=3$), **550.0 s** ($T_{he}=10$), and **955.3 s** ($T_{he}=20$). This shows that the dominant cost comes from $T_{he}$, and that small-to-moderate values provide the best practical trade-off. All experiments were conducted in PyTorch on an **RTX A6000 (47 GB)** GPU.

---

> > ### Author Rebuttal · Reviewer_uhTb · 2026-04-02
> >
> > Thanks for the response and new results. I will raise my score accordingly.

---

> > > ### Author Response · Authors · 2026-04-02
> > >
> > > We would like to thank the reviewer for recognizing the efforts made to address the concerns raised and for accepting our contribution to this important field.
> > >
> > > **Why Proposed Work Matters:** We believe the proposed optimization framework can provide the **inbuilt robustness** that deep neural networks currently lack during development. This fundamental flow is exploited by the adversaries, artificial or natural.
> > >
> > > The proposed research will help overcome this fundamental vulnerability.

---

### Official Review · Reviewer_jBb3 · 2026-03-13

**Soundness:** 3
**Presentation:** 3
**Significance:** 3
**Originality:** 3
**Overall Recommendation:** 4
**Confidence:** 3

**Summary:**

This paper explores fundamental challenge in deep learning: the vulnerability of neural networks to natural image corruptions. While much of the existing literature focuses on adversarial robustness, the robustness against common corruptions remains poorly understood. This paper attempts to explain this issue through an entropy-based analysis of neural network representations. The key idea comprise of analyzing the internal feature behavior of neural networks under corruptions and proposing a training strategy that improves corruption robustness by generating synthetic high-entropy samples.

**Compliance With Llm Reviewing Policy:**

Affirmed.

**Final Justification:**

The rebuttal effectively addresses the concerns I previously raised, particularly clarifying the identified weaknesses in the paper. Additionally, the new experimental results included in the rebuttal further strengthen the overall contribution and provide better support for the proposed method.

**Key Questions For Authors:**

How consistent is the observed entropy increase across different architectures and datasets?

**Limitations:**

yes

**Strengths And Weaknesses:**

Strengths

1. The paper introduces an interesting entropy-based explanation for corruption vulnerability. The empirical observation that corrupted samples produce higher entropy in deeper layers is insightful and helps explain performance degradation.

2. Dem-HEC proposed framework can be used for both CNN and Transformer architecture.

3. Experimental evaluations on multiple datasets including CIFAR-10, CIFAR-100, Tiny-ImageNet, and ImageNet-1K, with different architectures such as ResNet, RepVGG, ViT, and DINOv3 improve performance on common corruptions.


Weakness

1.  Comparison and difference between adversarial training and entropy maximization procedure would be helpful.

2. Why performance gains are marginal for ViT compared to CNN based architectures? Discussion on this would be insightful.

3. Only few corruption types are considered for evaluation, datasets like ImageNet-C include more corruptions.

4. Comparisons to works which focus on improving corruption robustness is missing [1,2].


[1] Apostolos Modas et al., PRIME: A Few Primitives Can Boost Robustness to Common Corruptions , ECCV 2022
[2] Chandramouli S. Sastry et al., DiffAug: A Diffuse-and-Denoise Augmentation for Training Robust Classifiers, NeurIPS 2024

---

> ### Author Rebuttal · Authors · 2026-03-31
>
> # 1 AT vs. EM
> Adversarial training works on the principle of bi-optimization, where inner optimization (maximization) aims to learn the worst perturbation $\delta$ and outer optimization  (minimization) ensures the robustness against it. It is observed that AT can help in achieving corruption robustness when it resembles small perturbations. Since this is not the case, the literature reports that AT provides only marginal robustness against common corruptions. Further, AT forces high confidence in correct labels for perturbed data, whereas EM encourages lower confidence (smoother outputs) on uncertain data. AT yields networks that are often poorly calibrated; at the same time, EM aims to improve calibration. Therefore, given the different goals of the two approaches (AT aims for correct prediction under worst-case perturbation, whereas the proposed approach aims for high uncertainty under perturbation), we assert that the proposed EM achieves better robustness than AT. To further showcase the strength of Dem-HEC, we also conducted experiments on **ImageNet-100 using a ResNet-18 backbone**. Under common-corruption evaluation, **Dem-HEC achieves 65.6**, which is **17.9 points higher than $l_\infty$ PGD AT (47.7)** and **17.2 points higher than $l_2$ AT (48.4)** on ImageNet100-C. For more details, please see our response to **Reviewer QBGh**.
> Another critical advantage of the proposed approach is that it reduces overconfidence by better uncertainty estimation. The literature shows that overconfidence is one of the primary factors in lower network robustness [1-2].
>
> [1] Robust models are less over-confident. NeurIPS'22
>
> [2] Mitigating neural network overconfidence with logit normalization. ICML'22
>
> # 2 ViT vs. CNN
> The smaller gains for ViTs compared to CNNs likely stem from the nature of **global self-attention**, which already encourages a more distributed and balanced representation across spatial tokens. As discussed in Section 5.2.2, this leaves **less room for improvement** relative to CNNs, even though Dem-HEC still gives consistent gains at higher severities (e.g., on Tiny-ImageNet, **Snow S5: 57.83 $\rightarrow$ 62.98** and **Pixelate S5: 59.78 $\rightarrow$ 64.11**).
>
> Importantly, this does not mean the method is limited to CNNs. In Section 5.2.3, we further show that Dem-HEC also scales to recent large models such as **DINOv3**, where it consistently improves corruption robustness on CIFAR100 (e.g., **Gaussian: 26.85 $\rightarrow$ 29.84**, **Shot: 33.48 $\rightarrow$ 36.19**, **JPEG: 49.56 $\rightarrow$ 52.29**). Thus, our key claim is not that ViTs improve more than CNNs, but that **Dem-HEC remains effective even for architectures and foundation-style models that already possess stronger inherent robustness**. We will clarify this discussion in the revised manuscript.
>
> # 3 Comparisons
>
> PRIME uses handcrafted augmentation primitives, DiffAug uses diffusion-based synthetic samples, and Dem-HEC improves robustness by generating bounded, high-entropy samples in representation space.
>
> While it is observed that the proposed approach does not always outperform existing benchmarks (though it is better than advanced benchmarks such as DiffAug), we believe our research provides new findings rather than only numerical advantages. We also conducted experiments to demonstrate that the proposed algorithm can complement existing approaches and be easily integrated with them. This combination can boost performance across all corruptions, including clean samples. For example, **DiffAug + Dem-HEC** improves all **15 corruptions**, with **+5.19** clean accuracy and **+5.98** average corruption accuracy over DiffAug.
>
> **CIFAR-10-C results with WideResNet-18-2**
> | Method | Clean | Gaus. | Shot | Imp. | Def. | Glass | Mot. | Zoom | Snow | Frost | Fog | Bright. | Contr. | Elast. | Pixel. | JPEG | Avg. |
> |:--|--:|--:|--:|--:|--:|--:|--:|--:|--:|--:|--:|--:|--:|--:|--:|--:|--:|
> | WideResNet | **81.20** | 76.84 | 77.53 | 76.57 | 68.75 | 67.14 | 59.84 | 64.12 | 72.95 | 70.08 | 64.34 | 78.08 | 48.30 | 68.94 | 69.71 | 76.19 | **69.29** |
> | Dem-HEC | **87.76** | 82.49 | 83.67 | 80.39 | 81.50 | 77.11 | 75.35 | 80.09 | 80.22 | 78.81 | 74.85 | 84.56 | 61.11 | 81.05 | 82.33 | 84.94 | **79.23** |
> | PRIME | 88.42 | 83.82 | 85.09 | 82.10 | 85.88 | 79.31 | 83.63 | 85.37 | 82.30 | 84.50 | 80.94 | 87.77 | 81.71 | 83.39 | 83.42 | 84.48 | **83.58** |
> | PRIME + Dem-HEC | **91.41** | 80.83 | 83.45 | 76.92 | 84.88 | 74.56 | 80.71 | 84.26 | **84.18** | **86.02** | 75.33 | **90.36** | 64.20 | **84.26** | **87.54** | **87.50** | 81.67 |
> | DiffAug | 84.14 | 79.85 | 80.62 | 79.35 | 76.92 | 71.24 | 66.13 | 73.90 | 75.36 | 74.77 | 69.99 | 81.62 | 51.59 | 75.41 | 76.97 | 80.37 | **74.27** |
> | DiffAug + Dem-HEC | 89.33 | **84.74** | **85.81** | **82.87** | 82.56 | 77.85 | 75.51 | 80.60 | 82.03 | 81.73 | 72.96 | 87.15 | 56.66 | 82.07 | 84.40 | 86.75 | **80.25** |
>
> # 4
>
> The entropy increase shows a consistent *trend* across architectures and datasets.

---

> > ### Author Rebuttal · Reviewer_jBb3 · 2026-04-03
> >
> > My concerns are resolved. I will increase score accordingly.

---

> > > ### Author Response · Authors · 2026-04-04
> > >
> > > We would like to thank the reviewer for recognizing the efforts made to address the concerns raised and for accepting our contribution to this important field.

---

### Decision · Program_Chairs · 2026-04-30

**Decision:**

Accept (regular)

**Comment:**

The paper identifies an interesting entropy-based explanation for why natural corruptions harm neural networks and proposes a mitigation. Across the discussion, reviewers viewed the core idea and empirical results positively. the AC therefore recommend accept